# A Method of Producing Low-Density, High-Strength Thin Cement Sheets: Pilot Run for a Glass-Free Solar Panel

**DOI:** 10.3390/ma16237500

**Published:** 2023-12-04

**Authors:** Jyh-Jeng Deng, Teng-Hsuan Lin, Jean-Shyan Wang, Yao-Chung Hsiao, Grung-Yi Tu, Qi-Hung Huang

**Affiliations:** 1Information Management Department, Da-Yeh University, Changhua 515006, Taiwan; 2Department of Environmental Engineering, Da-Yeh University, Changhua 515006, Taiwan; allenlin@mail.dyu.edu.tw; 3Department of Aeronautical Engineering, National Formosa University, Yunlin 632301, Taiwan; jswang01@nfu.edu.tw; 4Solarplant Technology, Taoyuan 325020, Taiwan; ych.solar@gmail.com; 5Department of Industrial Engineering and Management, Da-Yeh University, Changhua 515006, Taiwan; crungyi2356@gmail.com (G.-Y.T.); antony145236@gmail.com (Q.-H.H.)

**Keywords:** low-density and high-strength thin cement sheet, sandwiched cement sheet, EVA (ethylene vinyl acetate), backsheet, glass-free solar panel

## Abstract

This paper presents an innovative method of producing a low-density, high-strength, thin cement sheet. A seaweed powder was mixed with Portland cement, a foaming agent, calcium sulfoaluminate (CSA), and a quantity of water to create an A4-sized thin sheet with a thickness of 7 mm, which can withstand 1.5 kg in weight. This sheet was then covered with ethylene vinyl acetate and a backsheet to create a sandwiched cement sheet. The advantages of this sandwiched cement sheet are two-fold. First, it can support up to 13 kg in a static mechanical loading test, without bending, for over eight hours. Second, it can be quickly recovered at the end of its life cycle. This was a preliminary experiment to produce a large cement sheet that could satisfy the loading requirements for a solar panel. The purpose of the large, thin cement sheet is to replace the glass in a conventional solar panel and create a lightweight solar panel of less than 10 kg, which would mean that the installation of solar panels would become a one-person operation rather than a two-person operation. It would also increase the efficiency of the solar panel installation process.

## 1. Introduction

Solar panels convert sunlight to electricity based on the photovoltaic (PV) effect. However, there are several problems with solar PV panels, such as their low conversion efficiency and the rising concerns over PV waste after these panels are dismantled. In addition, solar panels in Taiwan have a mass of about 20 kg, meaning that it takes two people to install one on a roof. The majority of the weight of a solar panel is due to glass, which accounts for about 70% of the total weight of ordinary c-Si panels [1]. The density of glass is about 2.8 g/cm^3^; if it could be replaced with a lighter material, for example with a density of less than 0.9 g/cm^3^, the mass of a solar panel could be reduced from 20 kg to 10.5 kg. This would mean that the current two-person operation could be simplified to require only one person, which would significantly reduce labor costs and increase productivity. When two people install a 20 kg solar panel, coordination is required, whereas this coordination time is eliminated in a one-person operation. Glass has two main functions in a solar panel: the first is protection from damage, and the second is to allow sunlight to reach the solar cell. A further function of glass is to strengthen the solar panel, although the backsheet (which is usually made from Tedlar) also plays this role. A typical solar panel consists of six components: the frame, glass, two ethylene vinyl acetate (EVA) films (EVA#1 and EVA#2), the solar cells, and the backsheet. Note that the junction box is not considered here, as it is not a crucial element in the discussion. Figure 1 shows the various layers of a standard glass solar photovoltaic panel with an aluminum frame.

The six parts are arranged in the following way. Tempered glass is placed at the front of the PV module, as this is highly shock-resistant and can withstand even relatively large hailstones. This provides protection and transparency for solar cells. A plastic EVA film (EVA#1) is applied to the glass, and the interconnected PV cells are placed on it. Another EVA film (EVA#2) is then deposited on top of the PV cells. The EVA acts as an electrical insulator, preventing electrical contact between the front and back surfaces of the solar cells. Finally, the backsheet, which is usually made of polyvinyl fluoride (Tedlar), is laminated. The main function of the backsheet is to provide electrical insulation and protection for the solar cells, and it is usually colored white for a better reflectance of light [2]. The modules are framed and sealed with silicone sealant into aluminum profiles and connected to a junction box with output contacts. The junction box can be ignored in the function analysis, since it is not a matter of concern in this study.

Figure 2 shows the results of a function analysis of a solar panel. The parts of the solar panel are marked in blue, whereas the elements of the supersystem are marked in red. The frame supports the glass and protects the solar cells. The glass resists the elements (such as sand, hail, rain, and wind) and protects the EVA#1 layer. EVA#1 and #2 provide electrical insulation to the solar cells. The sun delivers photons to the solar cells, thereby providing electricity to appliances. The backsheet resists the elements, resists UV rays from the sun, and protects the EVA#2 layer.

From a simple function analysis of the solar panel, we can derive a glassless solar panel, as shown in Figure 3, Figure 4 and Figure 5 (from a Taiwanese patent [3]).

The transparent plate (marked 6) may be made of a translucent polymer plastic such as ethylene tetrafluoroethylene (ETFT), which is a high-quality fluoropolymer film. It is lightweight, has high light transmission and high durability, and can be used for surface protection for PV modules. The reflective plate (2) can be made of polyethylene terephthalate (PET); this reflects light so that the solar panels can capture more photons. The backsheet is mainly for electrical insulation and to reflect light away, so that the solar panel does not become too hot, as this would jeopardize electricity generation. There are two kinds of backsheets, soft and hard [4]. A soft backsheet may be made of polyethylene, polyamide, or PET film.

In contrast, a hard backsheet may be made of tempered glass, chemically strengthened glass, or synthetic resin. Since our proposed design is free of glass, the strength of the glass must be provided by a new backsheet, and, hence, only a hard backsheet is considered here. Initially, a bakelite board was considered for the backsheet; however, bakelite is expensive and may dissolve under high-temperature open conditions.

Further thought led to the choice of lightweight cement. If a lightweight cement sheet could be created with a thickness of 7 mm, a density of 0.9 g/cm^3^, and high strength, then it could serve as a backsheet to provide insulation and resistance to UV rays. The results of a function analysis of the new glass-free solar panel with a thin cement sheet as a backsheet is shown in Figure 6. Note that an important difference between Figure 2 and Figure 6 is that there are two additional objects in the function analysis of the glass-free solar panel: ETFT as a transparent plate and PET as a reflective plate. The function of the transparent ETFT plate is to resist elements and to protect EVA#1, whereas the function of the reflective PET plate is to reflect light to the solar cells. Note that the frame supports the thin cement sheet as a backsheet, meaning that the backsheet provides support for the whole solar module.

Although a cement sheet could easily be made with a density as low as 0.2 g/cm^3^ [5], its strength poses a problem. We, therefore, propose a solution for producing a thin, low-density cement sheet with high strength.

## 2. Literature Review

Glass-free solar panels, and particularly those based on monocrystalline silicon, have received a great deal of attention from practitioners in the area of solar power in terms of patent applications. Since silicon-based (c-Si) solar panels still dominate the market for PV panels (with an expected market share of up to 44.8% in 2030) [6], most current patents focus on how to create a lightweight (meaning glass-free) c-Si solar panel. US patent 2018/0309003 A1 [7] presented a solar panel consisting of solar cells, which included several supports to allow the panel to flex when strained while protecting the individual cells. For example, *depressions* can be defined as receiving individual cells with a spine or ridge between each cell to accept the pressure of foot traffic. The method mentioned in this patent used a special structure to eliminate the need for glass. Another patent took advantage of a special material called crosslink polymer: US patent 2010/0243033 A1 [8] described a method of manufacturing photovoltaic laminates in which a semiconductor material was sandwiched between layers of cross-linkable polymer material, which was then cross-linked. This method made it possible to avoid the use of adhesives (e.g., EVA), thereby eliminating the loss of incident sun energy from absorption by an adhesive layer. Suitable cross-linkable materials include thermo-setting polyester materials, polyurethane, polyacrylate, and epoxy resins. The term “epoxy resin”, as used here, generally refers to a cross-linked polymer of epichlorohydrin (chloromethyl oxirane) and bisphenol-A (4,4′-dihydroxy-2,2-diphenylpropane).

A third approach involved replacing glass with a sandwiched structural core layer sheet including two skins (made of e-glass fiber), a core (made of aramid honeycomb), an EVA layer, and an ETFE layer [9,10]. This structure had the advantage of combining them in a single process step. However, it was difficult to recycle due to the need to dismantle the sandwiched structures. To provide strength and avoid problems with recycling, Hsiao et al. [3] proposed a fourth method in Taiwan patent I769951, in which a bakelite board, or another substitute such as a thin cement sheet with a low density of 0.9 g/cm^3^ and high tensile strength, is required as a backsheet.

A lightweight and high-strength thin cement sheet would be highly desirable for the construction industry, as it could conserve space and reduce dead weight in buildings. A composite material is conventionally used for this purpose. For example, Wu et al. [11] developed a fiber-reinforced cement-based composite sheet for structural retrofits. Metakaolin was used with cement as the solid and was mixed with water; following this, 2D woven fabrics such as carbon and glass were used to make thin cement composite sheets. The dimensions of the standard sample were 76.2 (width) × 304.8 (length) × 5.0 mm (thickness). A 2D glass fiber has a high flexural strength, with a peak load of 70 MPa for a deflection of about 44 mm. In this report, only the density of glass and carbon fiber were considered, rather than the entire thin cement sheet. In view of the cement slurry process used, its density was expected to be greater than 2.3 g/cm^3^. In the Japanese patent no. JP S63-315206 [12], a manufacturing process for woven fabric-reinforced large-scale cement thin sheets was proposed. This method combined the oscillation molding of cement slurry with the use of reinforcing woven fabric whose pattern of the weave in a size of about 1.5–60 pieces/25 mm is arranged thereon. This process was simple and low-cost and could produce a large cement-based sheet with excellent properties in terms of toughness, crack resistance, and impact resistance.

A sample concrete sheet of a size of 1000 × 2000 × 5 mm was made with stainless steel fibers. It was observed that after steam-curing at 60 °C under 95% relative humidity (RH) for three days, no cracks or warpage were present. A specimen of a size of 100 × 30 × 5 mm was cut from the concrete sheet and subjected to a bending test, and the results show that the flexural strength was 211.8 kgf/cm^2^. Note that one kgf/cm^2^ equals 0.0980665 Mpa. However, a corresponding comparative test showed that many cracks were generated on the surface of the thin concrete sheet after one day of air curing, with these cracks occurring along the fibers. The density of this thin cement sheet was not reported.

Although the studies described above considered thin cement sheets, their densities of these sheets were regular rather than low. We, therefore, review some studies of low-density cement here. Lin et al. [5] proposed a composite material consisting of foam cement with a low density of 0.2 g/cm^3^ and containing a small amount of glass fiber as the filling material of a muffler to replace whole glass fibers. This new structure was referred to as a ‘green’ muffler, as it avoided the hazard of manual filling with chopped-strand fibers, and it was shown to give good noise reduction performance compared to a market muffler. In that paper, three cubes of a size of 200 mm were made with a density of 0.2 g/cm^3^, but the authors did not attempt to create a thin sheet of cement with a low density. Chen et al. [13] developed ultra-lightweight and high-strength engineered cementitious composites, with a density of about 1.23–1.26 g/cm^3^ and a corresponding compressive strength of 45 to 60 MPa, using 50 mm cubes, according to BS EN 12390–3:2009 [14].

From the literature review above, it is clear that there is a gap in the current research: there is a need for a low-density thin cement sheet with a high tensile strength. In the static mechanical load tests set out in IEC 61215-2:2016 [15], wind load is simulated by applying a load of ±2.4 kPa to a PV module, but there is no further description of how the test should be carried out. In their study of robust glass-free lightweight PV modules, Martins et al. [16] used 16-cell modules (size 810 × 810 mm) that were fixed using four clamps (width, 1.5 cm and length, 8 cm) placed at each corner when performing a static mechanical load test. Gabor et al. [17] used the LoadSpot tool for mechanical load testing, which was developed by BrightSpot Automation. This tool can perform standard static and cyclic load tests for panel certification, as per IEC-61215 [15] and IEC-DTS-62782 [18]. The aim of our study was to develop a formula for a low-density, thin cement sheet with an improved mechanical load testing capability which could be used as a backsheet to replace glass in an ordinary solar panel.

## 3. Materials, Methods, and Results

After a discussion with the client (a solar panel design company), we set out to make a thin cement sheet of a size of 29.0 × 20.0 × 0.7 cm that could bear a mass of 14.2 kg, corresponding to 2.4 kPa [15]. However, since the mass of 14.2 kg is expected to cover the whole surface of 29 × 20 cm, a widget weighing 7.1 kg (50% of the total mass) where the base covers only one-third of the whole surface of 29 × 20 cm can be used instead, as the bending effects of both can be shown to be approximately the same based on an ANSYS [19] analysis (see Appendix A). We, therefore, decided to search for a formula for a thin cement sheet that could withstand the required load of 7.1 kg over an area of 29.0 × 20.0 × 0.7 cm. It was also agreed that the thin cement sheet should be covered with an EVA layer of a thickness of 0.4 mm on both sides to strengthen the sheet. To take into account safety factors and for ease of calculation, the required load was increased to 7.5 kg for the cement sheet covered with EVA.

### 3.1. Initial Attempt 

An initial attempt was made to investigate the size of the load that a thin cement sheet could hold. A thin cement sheet with dimensions 20 × 20 × 1 cm was made with a density of 0.8 g/cm^3^. The corner broke as it was taken out of the mold, which had dimensions 20 × 20 × 20 cm. When a 200 g mass was placed on it, the sheet held, but it broke into pieces when a 300 g mass was applied. This showed that an ordinary thin sheet with a low density was too fragile to withstand the required load. The sheet is shown in Figure 7 and Figure 8. To increase the compressive strength of cement sheets, thin steel plates can be used as supports [20]; however, the thickness of a sheet of this type may reach 100 mm, which does not meet the requirement of a thin sheet. Another way to improve the strength is to stack panels of several materials, such as cement, reinforced steel, fiberglass, a geopolymer, and a carbon-fiber-reinforced polymer (CFRP). The thickness of these sheets exceeds 20 mm and may even be as high as 80 mm [21]. It also does not fulfill the requirement of a thin sheet. Finally, there is reactive powder concrete (RPC); a test concrete cylinder with a diameter of 50 mm and a length of 100 mm has been shown to have a compressive strength of as high as 120 MPa. Its formula included specific materials such as silica powder, quartz powder, steel fiber, cement, and sand [22]. However, this did not produce thin sheet samples either, and due to the complexity of the formula, this approach was not considered in our study.

In order to explore the relationship between the thickness and the load-bearing capacity of cement sheets, a progressive experimental approach was applied. Since a higher thickness gives a better load-bearing capacity, we made sheets of dimensions 30 × 30 × 2 cm, 30 × 30 × 1 cm, and 29.0 × 20.0 × 0.7 cm in sequence. If the load-bearing capacity of 7.5 kg was reached for the sheet of 30 × 30 × 2 cm, we then considered the next sheet of a size of 30 × 30 × 1 cm; if this sheet could withstand the load, we considered the next sheet of a size of 29.0 × 20.0 × 0.7 cm. Furthermore, since each sheet of different dimensions needed to be finished and the EVA glue needed to be applied before a load test of 7.5 kg was conducted at the client’s site, the process would have been time-consuming. Since the equipment used for glue coating was located at the client’s site, we decided to set a prerequisite for the experiment of 7.5 kg in the client’s laboratory: a load test of 1.5 kg in a university laboratory. This means that we put a prerequisite for the sheet in the university laboratory before it could be sent to the client for further load tests. The steps followed in this procedure are described in detail below.

### 3.2. Molds for Cement Sheet 

A two-layer wooden mold was developed that could be used to make cement sheets of 30 × 30 × 2 cm and 30 × 30 × 1 cm, as shown in Figure 9. If the movable wooden board was inserted, the dimensions of the sheet after grouting were 30 × 30 × 1 cm; otherwise, the dimensions were 30 × 30 × 2 cm. The results, after grouting, are illustrated in Figure 10. The other mold was an aluminum metal mold with a size of 29.0 × 20.0 × 0.7 cm, as shown in Figure 11.

A progressive experimental approach was applied as follows. We formulated a recipe for a 30 × 30 × 2 cm sheet, and if this was found to withstand a load of 1500 g, we used the same recipe for the 30 × 30 × 1 cm sheet. If this was also successful, we used it for the production of a thin sheet of a size of 29.0 × 20.0 × 0.7 cm. If one of these experiments was unsuccessful, we adjusted the formula, mainly by increasing the density of the cement and the amount of seaweed powder. The progressive experimental process is illustrated in Figure 12. The aim of this was to find an initial recipe for a thin cement sheet of a size of 29.0 × 20.0 × 0.7 cm. All of these experiments were performed in a university laboratory.

### 3.3. First Formula for Thin Cement Sheet with Seaweed Powder

One way to increase the density is to reduce the weight ratio of water to cement. Typically, the weight ratio of water to cement is 0.5, but, to increase the density, this can be reduced to 0.4. However, if the ratio is too low, a cement slurry will not form. A ratio that is too high is also unsuitable, since although the density will be reduced, the cement sheet will become brittle. To adjust the adhesion, the use of seaweed powder was considered [23]. In the construction industry, seaweed powder is mixed with water to form seaweed powder glue, which is then mixed with cement mortar and used as an adhesive to attach ceramic tiles to walls. The cement mortar used for this process may have a cement-to-sand ratio of 1:3 or may consist completely of cement. The weight ratio of seaweed powder to water used to form the seaweed powder glue is about 1:80–90, and the ratio of seaweed powder glue to cement mortar is 2:5. Finally, based on our experience of making foam cement silencers [5], our first formula for making foam cement containing seaweed powder was developed in accordance with Table 1.

The processes in Table 1 can be described in the following five steps: step 1, make seaweed glue; step 2, make cement slurry; step 3, make foam; step 4, make anhydrous calcium chloride solution; and step 5, make thin cement sheet. In step 1, we mixed 6 g of seaweed powder with 200 g of water 1 with a chopstick for five minutes to make a seaweed glue. To make the cement slurry in step 2, we used a ratio of Portland cement to calcium sulfoaluminate (CSA) of 9:1, based on recommendations given by Jones et al. [24]. This ratio was found to work very well in experiments with foam cement in silencers [5] and was, therefore, chosen for use in our experiments. The total mixtures in step 2 were 1080 g of Portland cement, 120 g of CSA, 206 g of seaweed glue (from step 1), and 600 g of water 2. They were mixed for five minutes with a spatula. Step 3 comprised foam making. The foaming agent in step 3 is a liquid, such as hydrogen peroxide and plant surfactants, which can generate foam when mixed with water. The foaming agent was about 5.5% of the total solution; thus, we used 13 g of the foaming agent and 225 g of water 3. Note that the foam solution weighed 238 g in total. The ratio of mixture 3 to mixture 2 was 238/2006 = 0.12. As the ratio decreased, the density of the foam concrete increased. Compared to the low density 0.2 g/cm^3^ cement slab in [5], where the corresponding ratio was 0.41, the ratio here was 0.12 for the thin cement sheet with a density of 0.6 g/cm^3^. Mixture 3 was made by a puddle mixer with a mixing time of seven minutes. In step 4, anhydrous calcium chloride and water 4 were mixed. Anhydrous calcium chloride was used to dehydrate the water in the foam concrete to harden it quickly. They were mixed with a chopstick for six minutes. Lastly, we put mixtures 2, 3, and 4 together in a bucket and mixed them for 15 min with a puddle mixer to make foam cement. Then, the foam cement was poured into the wooden mold to make the thin cement sheet.

Steps 1 and 2 are explained below. For 1200 g of cement (1080 g of Portland cement and 120 g of calcium sulphoaluminate), 600 g of water was required, as the weight ratio of water to cement for a cement mortar is 0.5. The formula for the seaweed powder glue in step 1 was the best value that was found. Recipes with 6 g of seaweed powder to 510 g of water and 50 g of seaweed powder to 960 g of water were tried, neither of which was good; the 30 × 30 × 1 cm cement sheet produced with the former recipe had a density of only 0.44 g/cm^3^ but was very brittle, whereas the 30 × 30 × 2 cm cement sheet made with the latter process had a density of 0.93 g/cm^3^ and was strong but expensive. The cost of a pack of 50 g of Nanshing-brand seaweed powder was NTD 50 (about USD 1.7).

The weight of the 30 × 30 × 1 cm cement sheet made according to the formula in Table 1 was 539 g, and the density was 0.6 g/cm^3^ (Figure 13). This sheet could bear a weight of 1500 g without breaking, as shown in Figure 14, where the two planks supporting the sheet are 15 cm apart. In the previous experiment, a cement sheet with a density of 0.8 g/cm^3^ and a thickness of 1 cm could carry less than 300 g. Due to the addition of seaweed powder, the resulting cement sheet has a lower density (0.6 g/cm^3^) but a much higher strength. It can be seen that the adhesive force produced by the seaweed powder was significant and improved the strength of the foam cement. Although a high-polymer mortar could also be used as an adhesive for tiles, the results of our tests with this material were not good.

The lower surface of the 30 × 30 × 1 cm cement sheet was carefully observed (the side formed at the bottom of the wooden mold). There were many fine holes on the surface that are not seen in traditional foamed cement, which has larger pores. The finer the holes, the stronger the foamed cement and, hence, the stronger the cement sheet. The reason for the formation of these fine holes seems to be due to the presence of seaweed powder, but the mechanism for this is currently unclear. According to [25], the seaweed powder contains algae, which can be integrated with Portland cement to form modified-polymer cement, which induces strong gelling and thickening capabilities, increasing the binding process. When used with cement, the algae powder will fill the porous portions and improve cement performance. These could be the reasons for the formation of these fine holes. The image on the left of Figure 15 shows the back of the sheet in Figure 13, and the image on the right shows the back of the sheet with the missing corner in Figure 7. Although both sheets are 1 cm thick, and the density of the sheet on the right (0.8 g/cm^3^) is higher than that on the left (0.6 g/cm^3^), the load capacity of the sheet on the left is higher. It should be noted that the pores in the sheet on the left are tiny but evenly and densely distributed on the surface, whereas, for the sheet on the right, they are mixed, with large and small pores that are unevenly distributed. Bearing in mind that a foam cement sheet of a 1 cm thickness and a density of 0.8 g/cm^3^ (without seaweed powder) can only hold less than 300 g in weight, a thin cement sheet of a 1 cm thickness and a density of 0.6 g/cm^3^ with seaweed powder can hold more than 1500 g in weight. This represents a five-fold increase in load-bearing capacity due to the addition of seaweed powder with the same thickness but a lower density.

A cement sheet of a size of 30 × 30 × 1 cm containing seaweed powder was taken to the client’s site for experiments. A layer of a 0.4 mm thick EVA glue was applied to the upper and lower surfaces of the sheet, and a layer of transparent film was then added. The experimental results are shown in Figure 16. From Figure 16a, it can be seen that the mass of the widget is 7650 g, while Figure 16b shows that the cement sheet undergoes no deformation under this load. Since a widget of a mass of 7.5 kg was not available, one of 7.65 kg was used instead. The experimental conditions in the industry is that the distance between the two supports supporting the cement sheet is 22 cm. Overall, this cement sheet was found to meet the requirements of the client.

### 3.4. Second Formula for Thin Cement Sheet with Seaweed Powder

Our next step was to make a sheet of a size of 29.0 × 20.0 × 0.7 cm. The first formula in Table 1 was found to be unsuitable, and after several failures, the formula in Table 1 was adjusted to increase its density. The results are summarized in Table 2. The water in step 2 was reduced from 600 g to 360 g, meaning that the weight ratio of water to cement was reduced to 0.3, which increased the density and hence the load-carrying capacity of the sheet. The results are shown in Figure 17. The mass of the thin cement sheet was 332 g, and the density was 0.81 g/cm^3^. It passed the 1.5 kg loading test, as shown in Figure 18. Figure 17 shows the front and back of the cement sheet; we note that the back surface contains bubbles, which are densely and finely distributed over the surface of the sheet.

The second cement sheet was sent to the client’s lab to be sealed with EVA before a 7.5 kg loading test, as shown in Figure 19. Its density was 0.84 g/cm^3^ after sealing, and it had the form of a sandwiched sheet. The purpose of the sandwich structure was to strengthen the load-bearing capacity. As can be seen from Figure 19a, the sheet was covered with EVA and a white backsheet, meaning that the front also appeared white. The back side, shown in Figure 19b, was covered with EVA and a transparent film made of PET. These materials are shown in Figure 20.

This time, a 2.6 kg widget was used for the loading test, as shown in Figure 21. The sandwiched cement sheet was laid on two wooden planks at a distance of 22 cm apart, as illustrated in Figure 22. Note that the unit in Figure 22 is mm. Two loading tests were performed: the first was a standard test, while the second was non-standard. In the first, three widgets with a total mass of 7.8 kg were placed on the sandwiched cement sheet, whereas in the second, five widgets with a total mass of 13 kg were used. The results are shown in Figure 23 and Figure 24, and the sheet passed both loading tests. We used a webcam to record the test from 7:42 p.m. on 5 October to 7:31 a.m. on 6 October. We note that in Figure 23, the load test of the sandwiched cement sheet lasts for more than three hours, and no deformation change occurs, whereas in Figure 24, the load test lasts for eight hours, and no deformation change occurs either.

These loading tests proved that the proposed sandwiched cement sheet with a size of 29.0 × 20.0 × 0.7 cm (original cement sheet dimensions) could bear a mass of up to 13 kg without breaking, far exceeding the design requirements of 7.5 kg.

## 4. Conclusions

PV energy is a renewable resource and can be used to meet the green energy requirements of current policies across the globe. However, the weight of the glass in a solar panel creates a barrier to its usage in buildings, particularly for installation on façades. Moreover, the heavy weight of solar panels impedes the efficiency of installation, as more labor is required. Several researchers have been working on the design and manufacture of glass-free solar panels. According to the function analysis of solar panels, there are two functions of glass: one is to support the panel and the other is to allow light to pass through. These two functions can be transferred to ETFT, a transparent plate to let the sunlight in and a thin cement sheet that can support the panel. To support the panel, it must withstand a 2400 Pa wind pressure, which poses a significant challenge.

One of the basic requirements for a solar panel is its ability to withstand wind pressures. After a discussion with the client, a solar panel design company, we agreed to make a thin cement sheet of 29.0 × 20.0 × 0.7 cm that could hold a weight of 14.2 kg, corresponding to 2.4 kPa [14]. We note that this weight covers the whole surface (29 × 20 cm), whereas, in reality, during a static mechanical loading test, the body of the widget does not usually cover the entire surface. Using ANSYS software (https://www.ansys.com/academic) [19], it can be shown that, when a widget with an area of about one-third of the whole surface is used, only half of the weight (7.1 kg) is required to achieve the same amount of bending. In this study, we aimed to find a formula for a thin cement sheet of a size of 29.0 × 20.0 × 0.7 cm that could support a widget of 7.5 kg. The reason for using 7.5 kg, rather than 7.1 kg, was simply for ease of calculation.

Researching the formula consisted of two steps. First, a formula had to be sought for a thin cement sheet of 30 × 30 × 1 cm. Using seaweed powder, a thin cement sheet of a size of 30 × 30 × 1 cm with a density of 0.6 g/cm^3^ was obtained, which passed the loading test with a widget of 7.6 kg. The formula for this sheet is shown in Table 1. The advantage of adding seaweed powder is that a foam cement sheet of a 1 cm thickness and a density of 0.8 g/cm^3^ (without seaweed powder) can only hold less than 300 g in weight. In contrast, a thin cement sheet of a 1 cm thickness and a density of 0.6 g/cm^3^ with seaweed powder can hold more than 1500 g in weight. The load-bearing capacity is increased five-fold due to the addition of seaweed powder, with the same thickness but a lower density. According to [22], seaweed powder contains algae, which can be integrated with Portland cement to form modified-polymer cement, and this induces strong gelling and thickening capabilities, increasing the binding process. When used with cement, the algae powder will fill the porous portions and improve cement performance.

The formula was then modified, as shown in Table 2, to produce a thin cement sheet of a size of 29.0 × 20.0 × 0.7 cm, with a density of 0.81 g/cm^3^. The water in step 2 was reduced from 600 g to 360 g, meaning that the weight ratio of water to cement was reduced to 0.3, which increased the density and, hence, the load-carrying capacity of the sheet.

The thin cement sheet was then covered with EVA and a backsheet to create a sandwiched cement sheet and to enhance its load-bearing capacity. The density of the sandwiched cement sheet was 0.84 g/cm^3^, and it passed loading tests of 7.8 kg and 13 kg, respectively. According to the ANSYS analysis, only the loading test of 7.5 kg was required. However, in order to show the excellence of the load bearing capacity of the sandwiched cement sheet, two loading tests were performed. The first was a standard test while the second was non-standard. In the first test, three widgets with a total mass of 7.8 kg were placed on the sandwiched cement sheet. In the second test, five widgets with a total mass of 13 kg were used. The results are shown in Figure 23 and Figure 24, and the sheet passed both loading tests. We used a webcam to record the test from 7:42 p.m. on 5 October to 7:31 a.m. on 6 October. Figure 23 shows that the load test of the sandwiched cement sheet lasted for more than three hours, and no deformation change occurred, whereas, in Figure 24, the load test lasted for eight hours, and no deformation change occurred there either.

In the future, solar panels will need to be made with sheets larger than 29.0 × 20.0 × 0.7 cm [14]. Furthermore, the electricity power output will need to be investigated after a static mechanical loading test. However, as a pioneer run, the research results in this paper will pave the way towards this final goal. It should also be noted that the results of this research could also be applied in other areas of construction, such as façade installation, both outside and inside a building.

## Figures and Tables

**Figure 1 materials-16-07500-f001:**
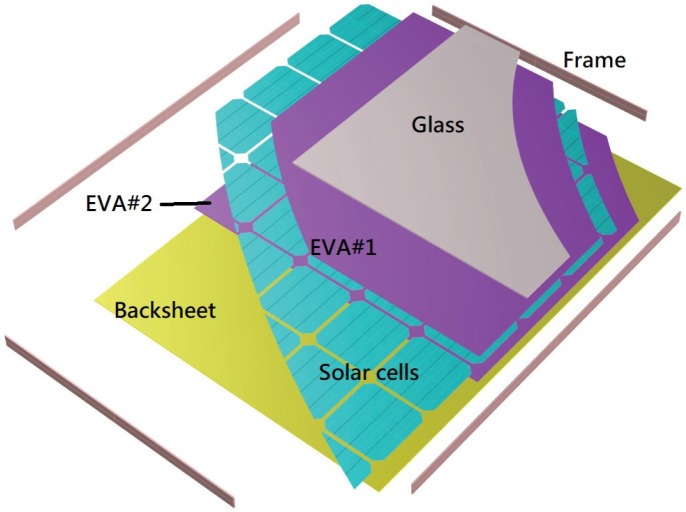
Structure of a photovoltaic module.

**Figure 2 materials-16-07500-f002:**
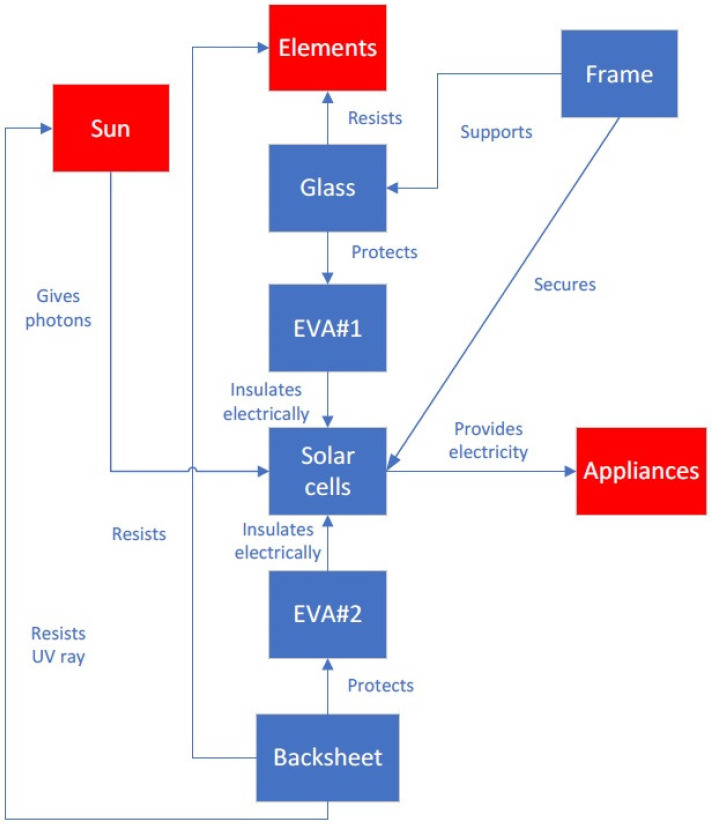
Results of a function analysis of a solar panel.

**Figure 3 materials-16-07500-f003:**
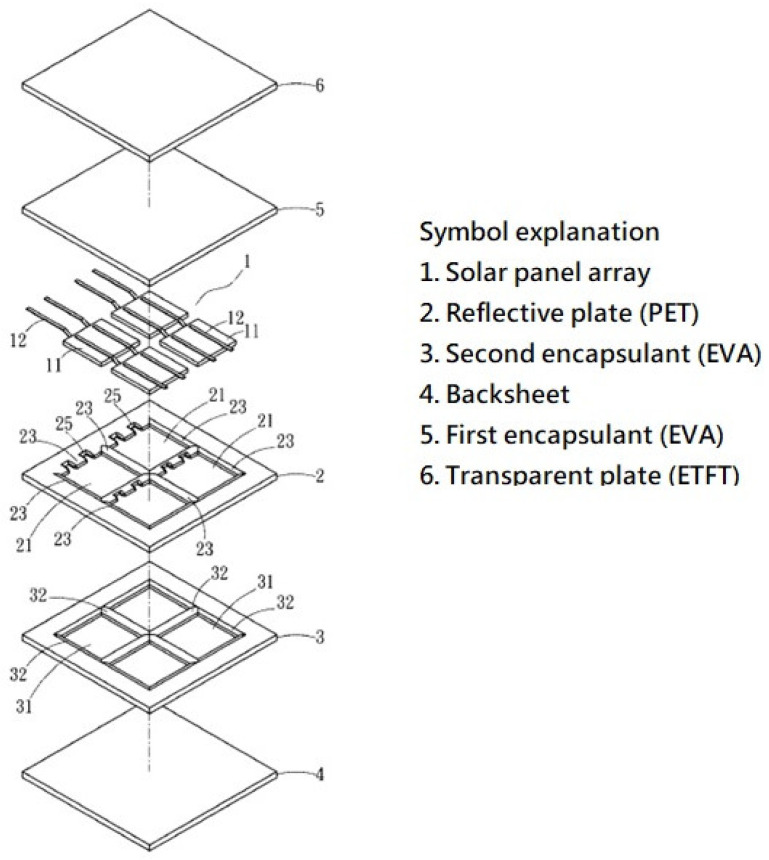
Exploded view of a glassless panel [3].

**Figure 4 materials-16-07500-f004:**
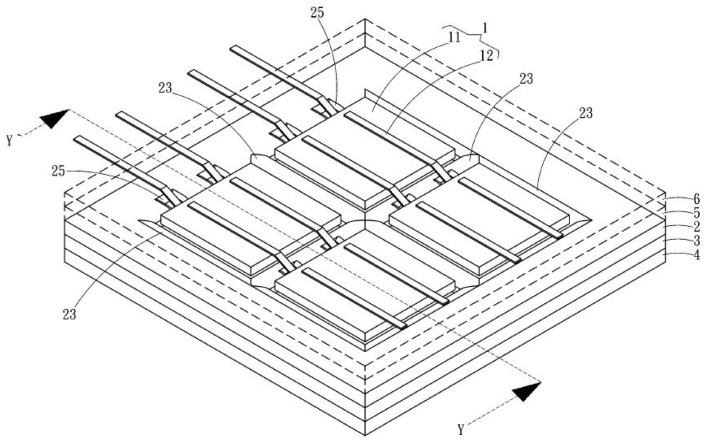
Perspective view of a glassless panel [3].

**Figure 5 materials-16-07500-f005:**
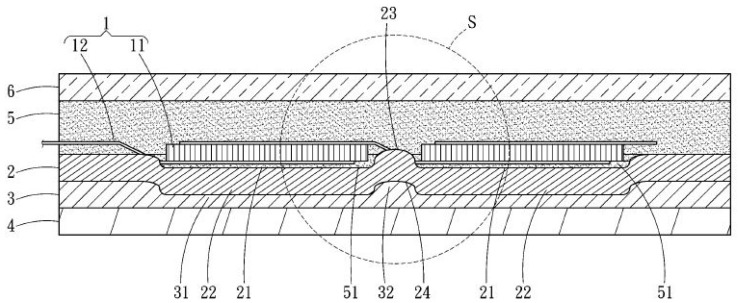
Section view of a glassless panel along the Y–Y′ direction [3].

**Figure 6 materials-16-07500-f006:**
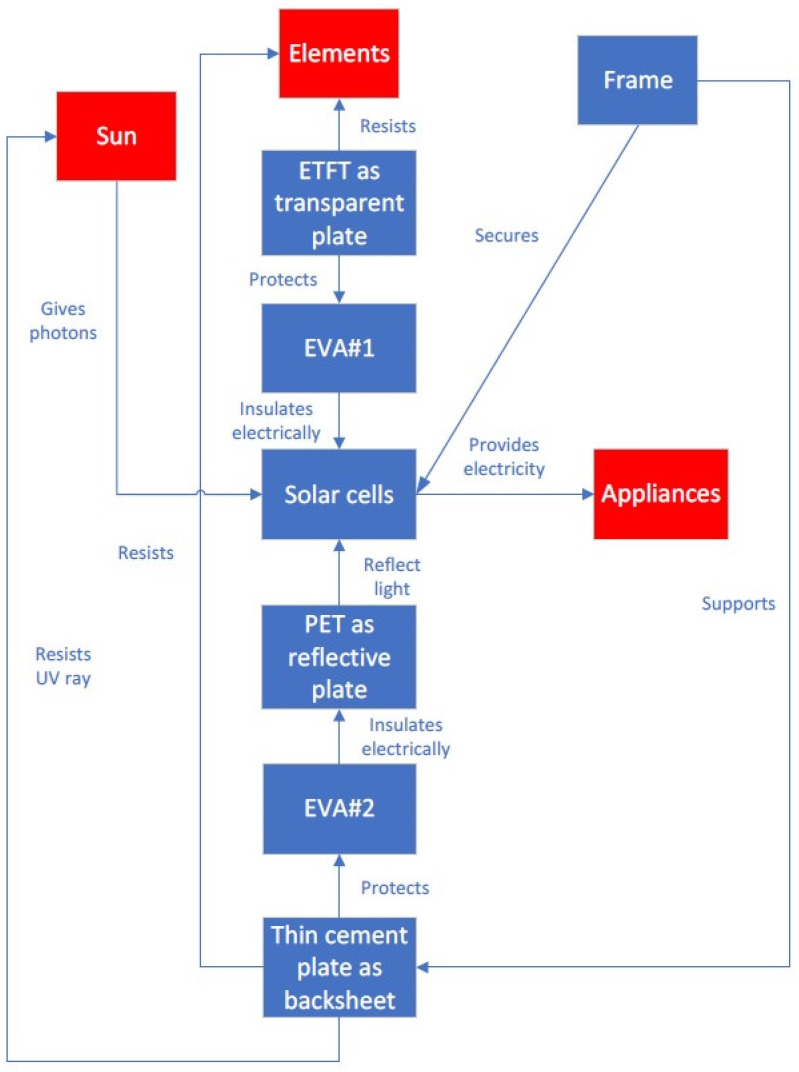
Results of a function analysis of a glass-free solar panel.

**Figure 7 materials-16-07500-f007:**
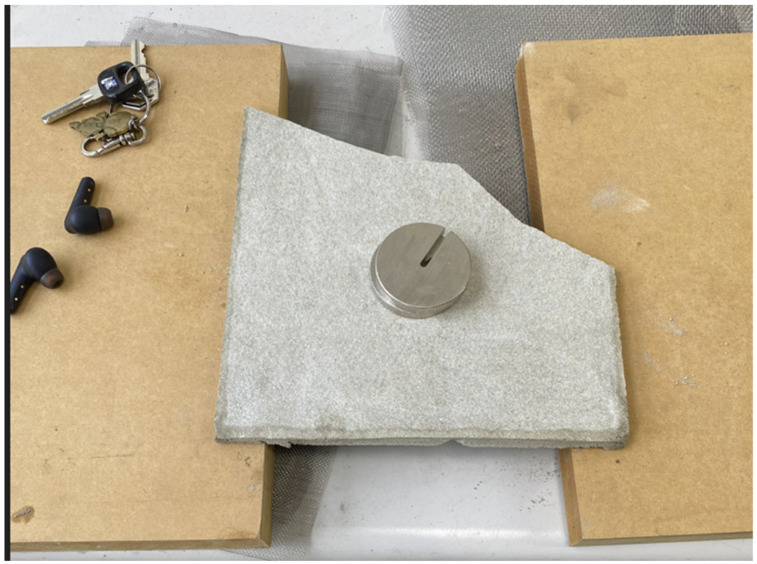
Thin cement sheet with a 200 g mass.

**Figure 8 materials-16-07500-f008:**
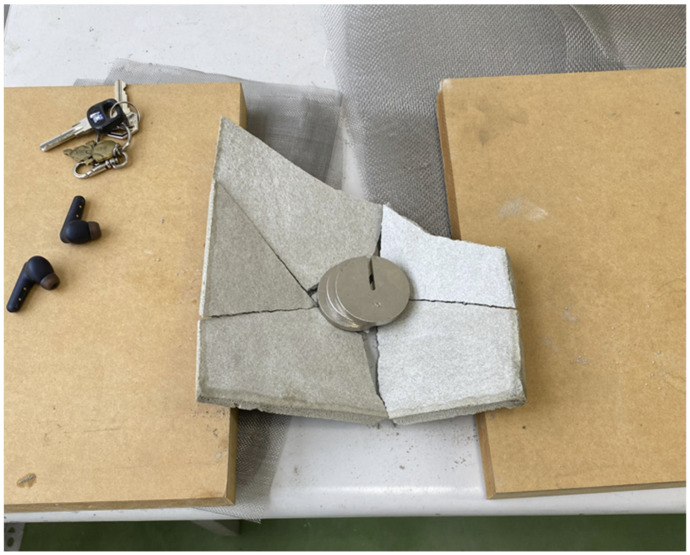
Thin cement sheet with a 300 g mass.

**Figure 9 materials-16-07500-f009:**
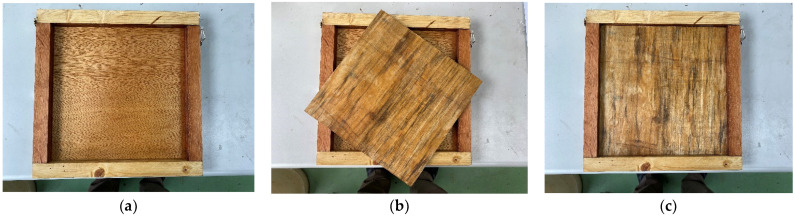
Two-layer wooden mold. (**a**) Mold with a depth of 2 cm; (**b**) Mold + 1 cm thick board; (**c**) Mold with a depth of 1 cm.

**Figure 10 materials-16-07500-f010:**
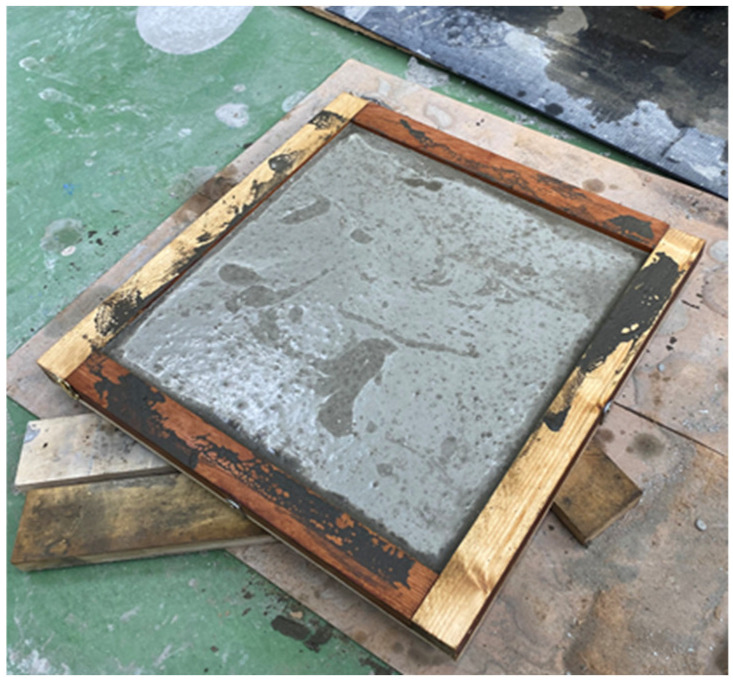
Wooden mold after grouting.

**Figure 11 materials-16-07500-f011:**
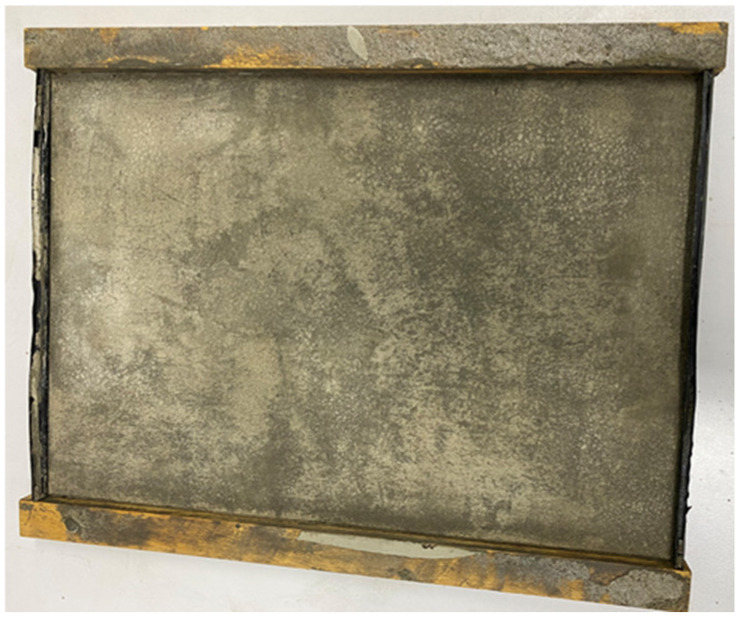
Metal mold with a depth of 7 mm.

**Figure 12 materials-16-07500-f012:**
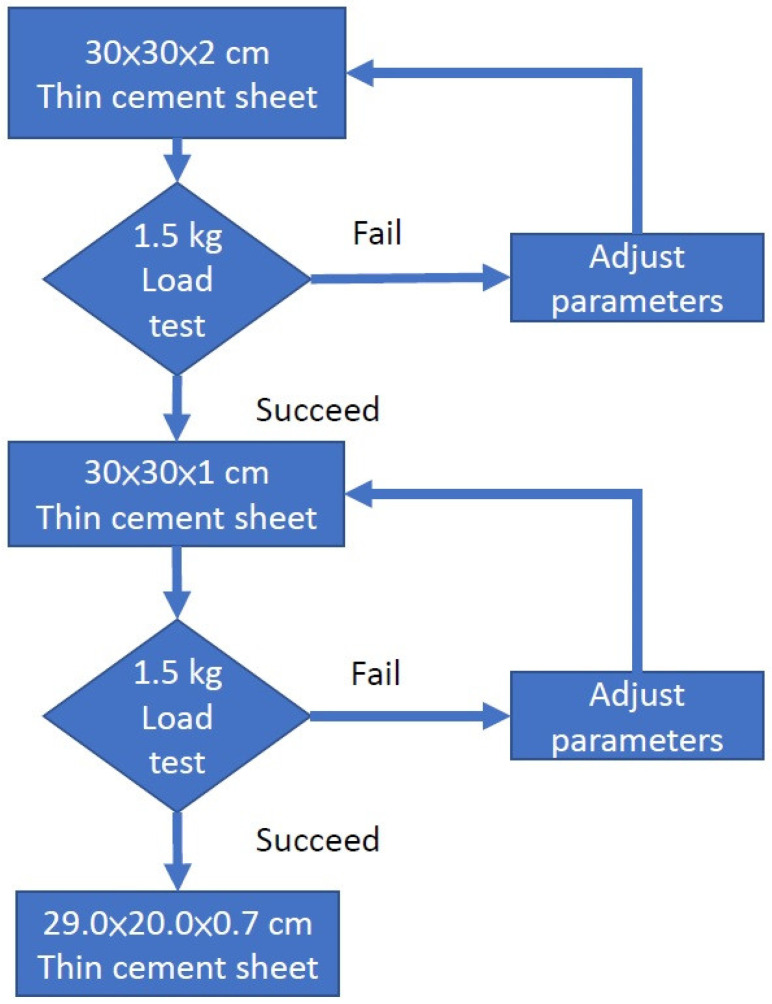
Progressive experimental process.

**Figure 13 materials-16-07500-f013:**
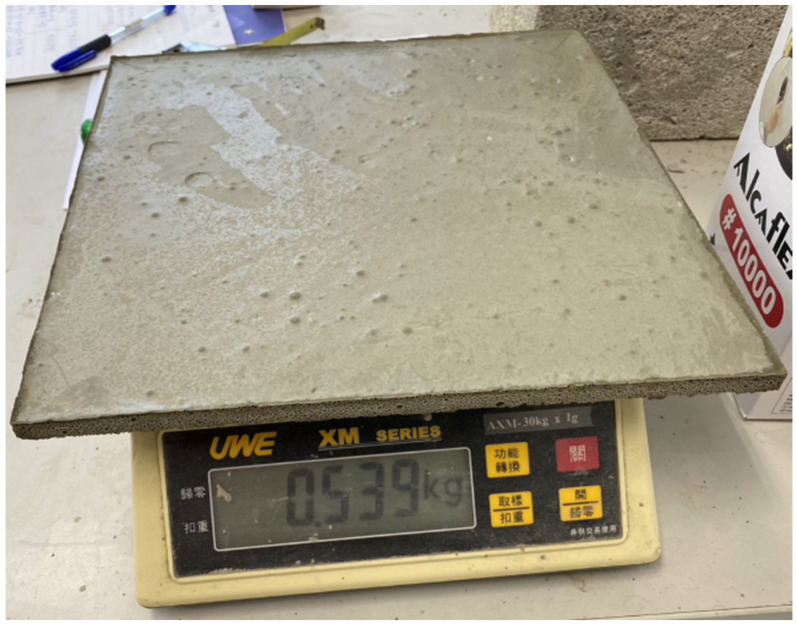
A 30 × 30 × 1 cm cement sheet of a mass of 539 g.

**Figure 14 materials-16-07500-f014:**
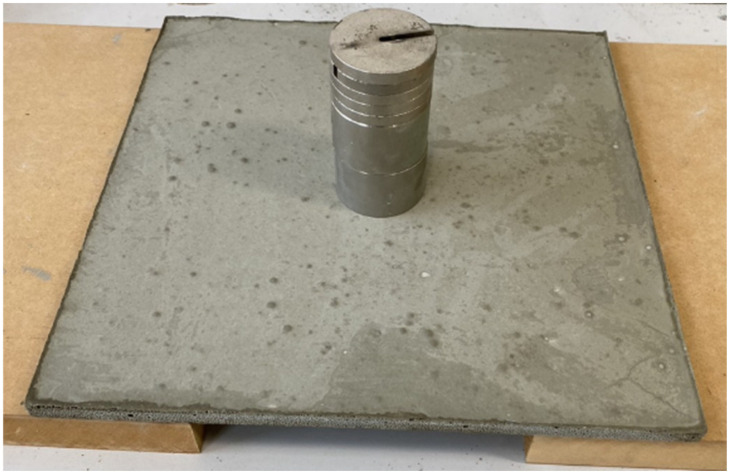
First cement sheet bearing a mass of 1500 g.

**Figure 15 materials-16-07500-f015:**
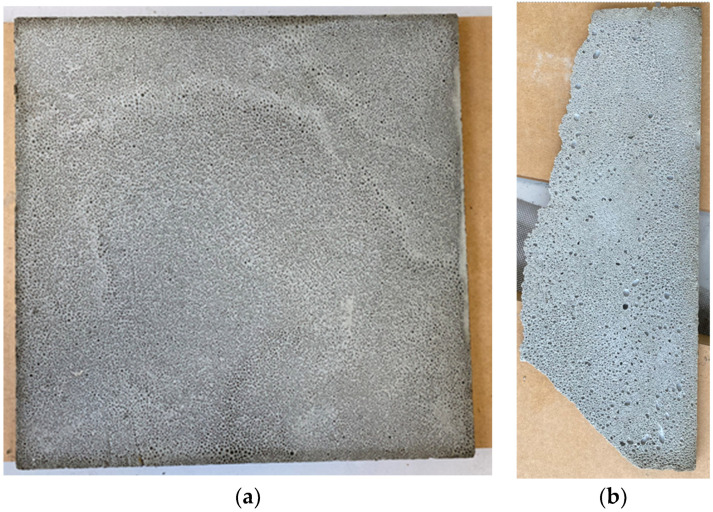
Comparison between the backs of two cement sheets. (**a**) Back of the strong cement sheet; (**b**) Back of the weak cement sheet.

**Figure 16 materials-16-07500-f016:**
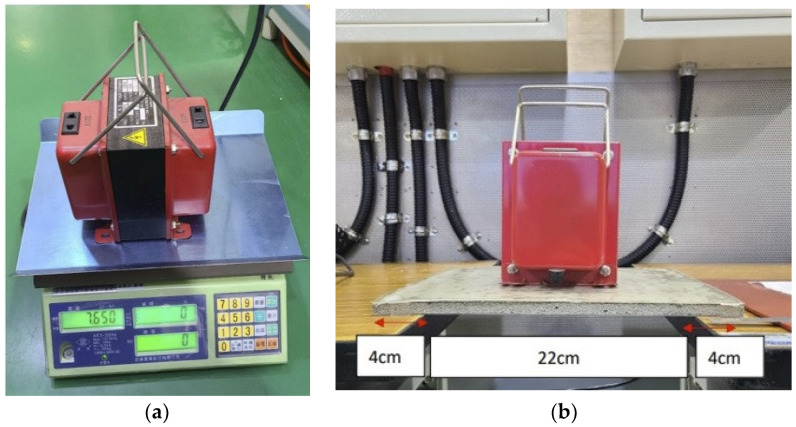
Setup used for the 7.5 kg loading test in the client’s laboratory. (**a**) Widget weight; (**b**) Setup used for the loading test.

**Figure 17 materials-16-07500-f017:**
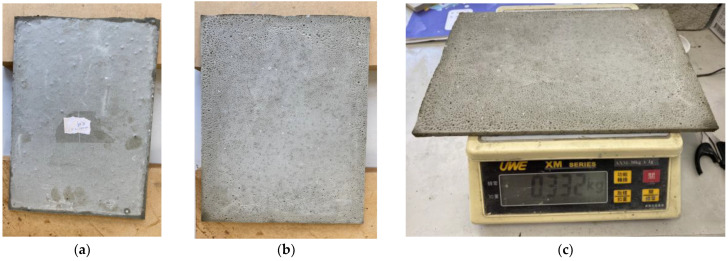
Second cement sheet. (**a**) Front; (**b**) Back; (**c**) Sheet on the scale.

**Figure 18 materials-16-07500-f018:**
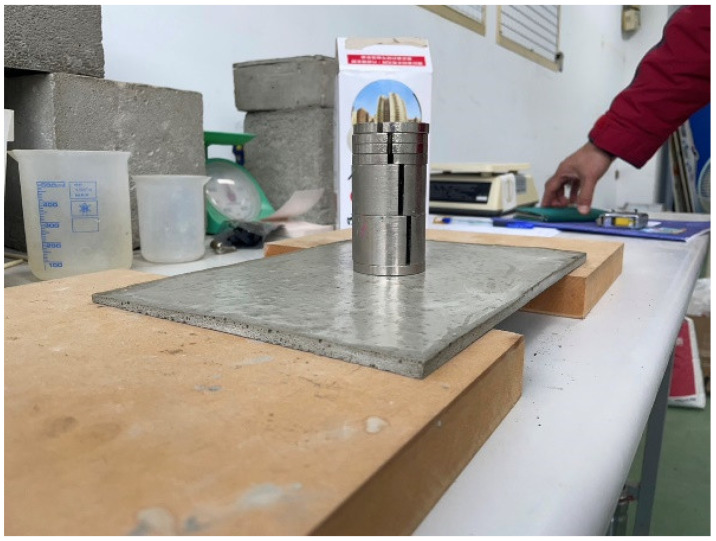
Second cement sheet supporting a mass of 1500 g.

**Figure 19 materials-16-07500-f019:**
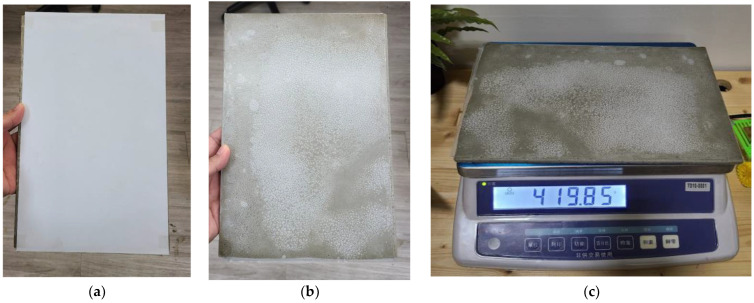
Second cement sheet after sealing. (**a**) Front; (**b**) Back; (**c**) Sealed sheet on the scale.

**Figure 20 materials-16-07500-f020:**
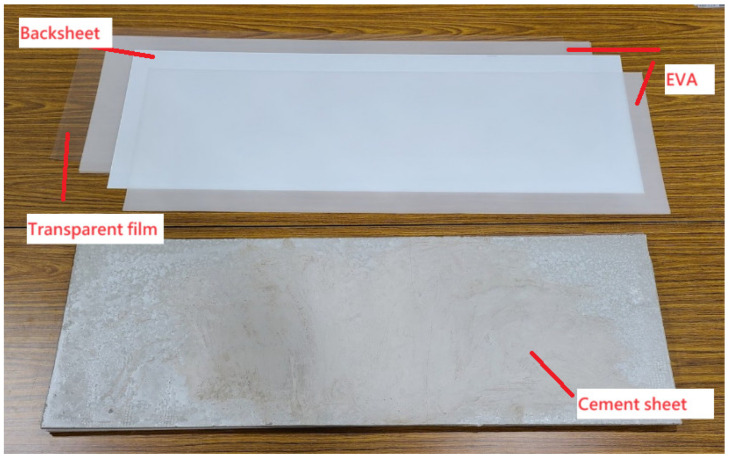
Materials used for the sandwiched cement sheet.

**Figure 21 materials-16-07500-f021:**
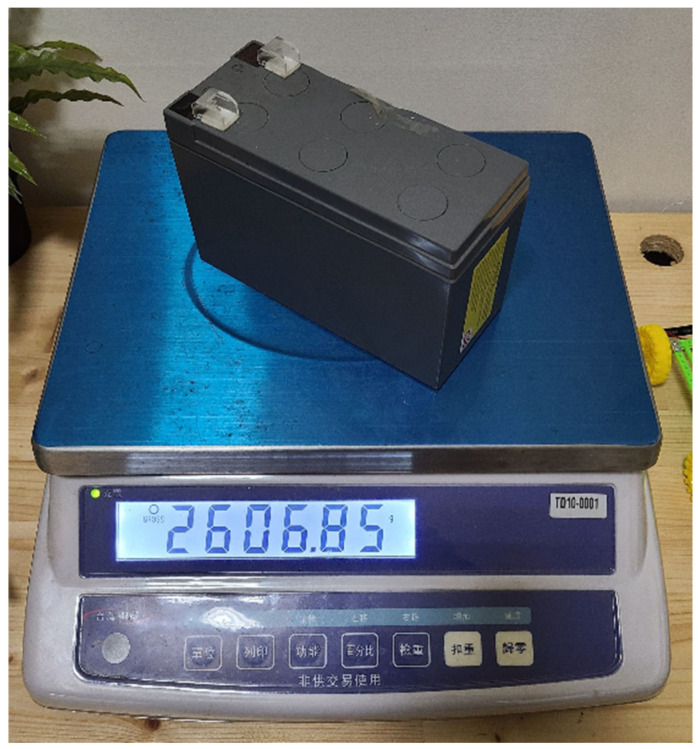
A widget of 2.6 kg.

**Figure 22 materials-16-07500-f022:**
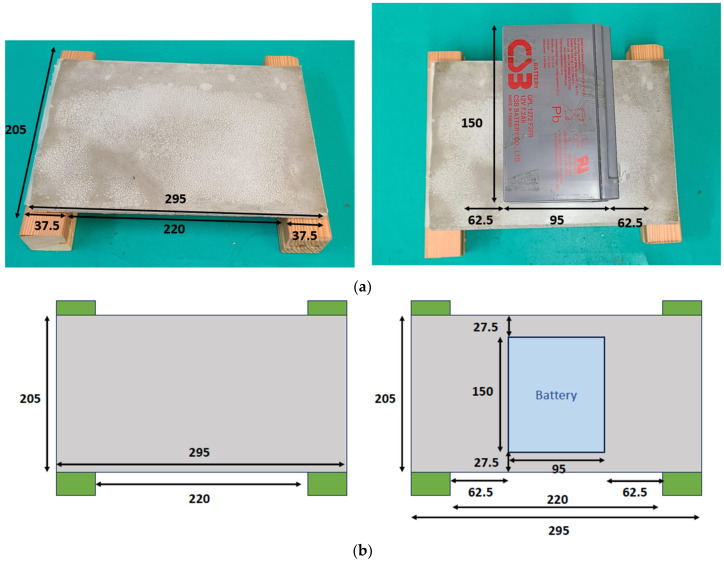
Sandwiched cement sheet on wooden planks. (**a**) Artifact layout; (**b**) Drawing layout.

**Figure 23 materials-16-07500-f023:**
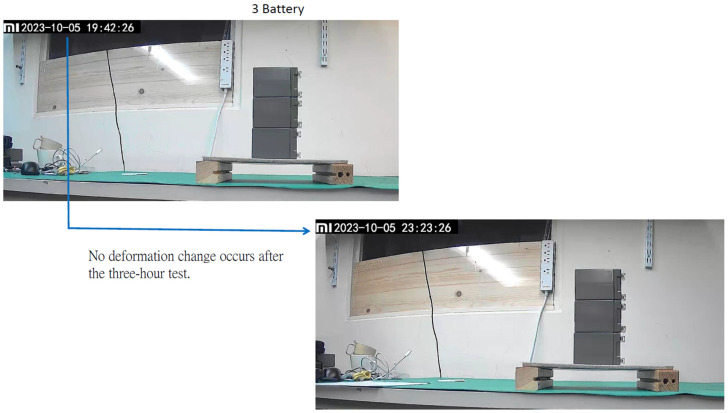
Loading test of a sandwiched cement sheet with three widgets.

**Figure 24 materials-16-07500-f024:**
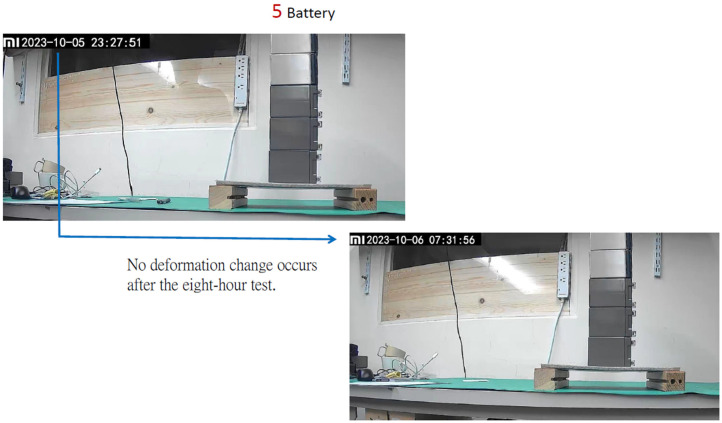
Loading test of a sandwiched cement sheet with five widgets.

**Table 1 materials-16-07500-t001:** First formula for foam cement.

Production Procedure	Material	Weight (g)	Mix Time (min)
1. Seaweed glue	Seaweed powder	6	5
	Water 1	200	
2. Cement slurry	Portland cement	1080	5
	Calcium sulfoaluminate (CSA)	120	
	Seaweed glue (step 1)	206	
	Water 2	600	
3. Foam	Foaming agent	13	7
	Water 3	225	
4. Calcium chloride anhydrous solution	Calcium chloride anhydrous	30	6
	Water 4	26	
5. Foam cement	Cement slurry (step 2)	2006	15
	Foam (step 3)	238	
	Calcium chloride anhydrous solution (step 4)	56	

**Table 2 materials-16-07500-t002:** Second formula for foam cement.

Production Procedure	Material	Weight (g)	Mix Time (min)
1. Seaweed glue	Seaweed powder	6	5
	Water 1	200	
2. Cement slurry	Portland cement	1080	5
	Calcium sulfoaluminate (CSA)	120	
	Seaweed glue	206	
	Water 2	360	
3. Foam	Foaming agent	13	7
	Water 3	225	
4. Calcium chloride anhydrous solution	Calcium chloride anhydrous	30	6
	Water 4	26	
5. Foam cement	Cement slurry (step 2)	1766	15
	Foam (step 3)	238	
	Calcium chloride anhydrous solution (step 4)	56	

## Data Availability

Data are contained within the article.

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
