# Peer review of "A Method of Producing Low-Density, High-Strength Thin Cement Sheets: Pilot Run for a Glass-Free Solar Panel"

_materials, 2023, doi:10.3390/ma16237500_

Round 1

Reviewer 1 Report

Comments and Suggestions for Authors

The comments and suggestions are attached for minor corrections. After the minor correction (if possible), the article shall be re-submitted for publication.

Conclusion: Accept for publication.

Reviewer 2 Report

Comments and Suggestions for Authors

The authors describe their work to develop low density cement sheets that can resist a bending load.

The results are clearly important for the application and the characterization tests highly targeted to this effect.

The article is clearly written.

In terms of material development, the foaming process and general sample preparation are not described. The microstructure is not studied, except by qualitative appreciation of “tiny” versus “large” pores. The mechanical properties (3-point bending strength, compressive strength, etc) are not described. All these points make the work less valuable for the scientific community targeted as an audience by this journal.

Reviewer 3 Report

Comments and Suggestions for Authors

A Method of Producing Low-Density, High-Strength Thin Cement Sheets

The article is interesting and well written. A few suggestions are provided below to further improve the quality of the article.

1.                           Abstract need revision with some quantitative results.

2.                           Some more latest studies are required in the background section to further highlight the background and necessity of this study. Please consider the following as well.

2.1.                     Jusoh, S.  

2.2.                     N., Mohamad, H., Marto, A., Yunus, N. M., & Kasim, F. (2015). Segment’s joint in precast tunnel lining design. J Teknol, 77(11), 91-98.

2.3.                     Namazi, E., Mohamad, H., Jorat, M. E., & Hajihassani, M. (2011). Investigation on the effects of twin tunnel excavations beneath a road underpass. Electronic Journal of Geotechnical Engineering, 16(1), 1-8.

2.4.                     Hisham, M., Peter, J., Kenichi, S., Assaf, K., & Adam, P. (2007). Distributed optical fiber strain sensing in a secant piled wall. In Proceedings of the Seventh International Symposium on Filed Measurements in Geomechanics (Vol. 1, pp. 22-31). sn.

3.                           Authors must summarize results in a more systematic way with reference to the previous studies.

4.                           The section mentions the use of seaweed powder to enhance adhesion and strength in the cement sheets. However, it lacks a detailed explanation of how seaweed powder functions in this context. More information on the chemical or physical properties of seaweed powder and its interaction with the cement would be helpful.

5.                           The section notes that the addition of seaweed powder results in finer holes in the cement sheets. While this is considered advantageous, the mechanism behind the formation of these fine holes is left unexplained. A brief discussion of why and how seaweed powder affects the pore structure of the sheets would add depth to the analysis.

6.                           Also, Conclusions are too limited to prove the significant outcome of this study. The conclusions should be improved.

Comments on the Quality of English Language

Minor editing of english language required.

Reviewer 4 Report

Comments and Suggestions for Authors

A Method of Producing Low-Density, High-Strength Thin Cement Sheets

Authors: Jyhjeng Deng1* , Teng-Hsuan Lin2 , Jean-Shyan, Wang3 , Y.C. Hsiao4 , Grung-Yi Tu5 and Qi-Hung Huang5

Reviewer’s comments

General comment: the work is original from the practical point of view and specifically for readers working with lightweight materials. The authors place the greatest emphasis on the choice of material composition. This may be of interest only to a very narrow range of readers. The overall merit to the material science is very low.

Comments in detail:

Introduction and literature review sections are coherent and provide a reasonable explanation on the purpose of the research.

Experimental and Results

Authors used one term – a cement sheet in introduction section however, later, a term foamed concrete appeared. I suggest authors to use a single terminology throughout the paper.

Line 196 – the application of ANSYS software. I am curious why the authors used an aluminum plate in ANSYS calculation? The aim was to produce a porous lightweight material and although aluminum is lightweight, it is a homogeneous material without pores. Although a simplified attempt was useful to you, it is more correct to use the similar material  to simulate a respond.

Line 276-277 – it is not clear what kind of material authors produce – “low-density foamed cement” or “foamed concrete”. These two terms should be separated or clear explanation of their use should be provided.

Line 277-290 – a “step 1” is missing in Table 1. It is not clear, when you added the calcium chloride and whether you add the seaweed glue during production of the final foamed concrete? Please clarify the text.

Line 296, 298, 306 – once again, new terms are used - cement board and foamed cement. Is it a foamed concrete or just a board produced from cement slurry? Text needs explanations.

Line 365-382 Authors applied a static load only, whereas the dynamic or cyclic loads are important for the backsheets. I believe that authors will consider the possibility of conducting experiments on the resistance of sheet to dynamic or cyclic loads.

Conclusions 

This is the weakest part and needs to be revised. The authors should present their results rather than repeating the text provided in the introduction and experimental sections. In my opinion, the text presented in lines 385-392 is completely unnecessary.

Round 2

Reviewer 2 Report

Comments and Suggestions for Authors

The authors describe their work to develop low density cement sheets that can resist a bending load.

The results are clearly important for the application and the characterization tests highly targeted to this effect.

The article is mostly clearly written (except new edits), in terms of English, but does not comply to scientific standards. Examples are given below.

In terms of mechanical properties, some calculation of the tensile strength should be attempted assuming linear elasticity and a 4-point bending load.

The article mentions “client” several times but the funding is indicated as “not applicable”. The instructions indicate “Funding: All sources of funding of the study should be disclosed.

-----------------------------------------------------------

L14: New confusing edits in abstract:” This thin cement sheet was developed in two steps: the 1st first formula for foam cement and the second formula for foam cement.”: ??

L16: Confusing text in abstract: “In the first step, a seaweed powder was added to the regular foam cement to make a 30 × 30 × 1 cm cement sheet with a 0.6 g/cm3 density. This sheet could bear a weight of 1500 g without breaking. A cement sheet with a density of 0.8 g/cm3 and a thickness of 1 cm can carry less than 300 g.”

L 27: The following sentence is not backed by data: “our method is much more environmentally friendly.”

L166: unit kgf/cm2

L214: “Brittleness” : Brittle opposes ductile. Lack of scientific precision in wording.

L214-224: The paragraph is confusing.

L229-244: No chronology is needed. Only assumptions, results and discussion.

Fig. 9-11: needed?

L261 “we adjusted the formula, mainly by increasing the density and adhesion of the cement.”. What does increase adhesion mean? Is adhesion measured/quantified? Specific information would be “the amount of x was increased in the recipe, targeting an increase in cement density and xxx”

Table 1: What is the foaming agent?

Fig. 17 (c) and 19 (c ) , Fig. 21: Needed?

L 413: “without bending” would mean in infinitively stiff plate. Are authors meaning without breaking?

Comments on the Quality of English Language

I have no further comment.

Author Response

We highly appreciate the comments of the reviewer and have considered them carefully. In response, we have deleted part of the paragraph in the abstract and added new contents in red.

  1. The authors describe their work to develop low density cement sheets that can resist a bending load.

    1A. Thank you for the positive comments.

  2.  The results are clearly important for the application and the characterization tests highly targeted to this effect.

    2A. Thank you for the positive comments.

  3. The article is mostly clearly written (except new edits), in terms of English, but does not comply to scientific standards. Examples are given below.

    3A. Thank you for the constructive comments. Corrections are made accordingly.

  4.  In terms of mechanical properties, some calculation of the tensile strength should be attempted assuming linear elasticity and a 4-point bending load.

    4A. Thank you for your suggestion. One must damage the sandwiched cement sheet to calculate the tensile strength via a 4-point bending load. Due to the hard labor of making a sandwiched cement sheet, we want to ensure the sample is intact. Thus, tensile strength will not be reported in this paper.

  5. The article mentions “client” several times but the funding is indicated as “not applicable”. The instructions indicate “Funding: All sources of funding of the study should be disclosed.

    5A. Thank you for your suggestion. The funding has been changed to “Funding: Part of the research funding comes from Solarplant Technology, Taiwan.” It is shown in Line 463 in the revision paper.

  6. L14: New confusing edits in abstract:” This thin cement sheet was developed in two steps: the 1st first formula for foam cement and the second formula for foam cement.”: ??

    6A. Thank you for your suggestion. This confusing statement is deleted from the abstract. Indeed, to make it concise, the abstract has been completely modified. Part of it is repeated for your reference.

    This paper presents an innovative method of producing a low-density, high-strength thin cement sheet. A seaweed powder was mixed with Portland cement, a foaming agent, calcium sulfoaluminate (CSA), and a quantity of water to create an A4-sized thin sheet with a thickness of 7 mm, which can withstand 1.5 kg in weight. This sheet was then covered with ethylene vinyl acetate and a backsheet, to create a sandwiched cement sheet. The advantages of this sandwiched cement sheet are two-fold. First, it can support up to 13 kg in a static mechanical loading test, without bending, for over eight hours. Second, it can be quickly recovered at the end of its life cycle. …

  7. L16: Confusing text in abstract: “In the first step, a seaweed powder was added to the regular foam cement to make a 30 × 30 × 1 cm cement sheet with a 0.6 g/cm3 density. This sheet could bear a weight of 1500 g without breaking. A cement sheet with a density of 0.8 g/cm3 and a thickness of 1 cm can carry less than 300 g.”

    7A. The answer is the same as that of 6A.

  8. L 27: The following sentence is not backed by data: “our method is much more environmentally friendly.”

    8A. The answer is the same as that of 6A.

  9. L166: unit kgf/cm2

    9A. We add one sentence to explain kgf/cm2 as “Note that one kgf/cm2 equals 0.0980665 Mpa” in Line 156.

  10. L214: “Brittleness” : Brittle opposes ductile. Lack of scientific precision in wording.

    10A. Thank you for your suggestion. Brittleness is deleted. The original sentence is changed to “To increase the compressive strength of cement sheets, thin steel plates can be used as supports [17]” to make it clear in Line 204.

  11. L214-224: The paragraph is confusing.

    11A. Thank you for your suggestion. The whole paragraph is completely written in Lines 203-215. They are repeated here.

    To increase the compressive strength of cement sheets, thin steel plates can be used as supports [17]; however, the thickness of a sheet of this type may reach 100 mm, which does not meet the requirement of a thin sheet. Another way to improve the strength is to stack panels of several materials, such as cement, reinforced steel, fiberglass, geopolymer, and carbon-fiber-reinforced polymer (CFRP). The thickness of these sheets exceeds 20 mm and may even be as high as 80 mm [18]. Neither does it fulfill the requirement of a thin sheet. Finally, there is reactive powder concrete (RPC); a test concrete cylinder with a diameter of 50 mm and a length of 100 mm has been shown to have a compressive strength of as high as 120 MPa. Its formula included specific materials such as silica powder, quartz powder, steel fiber, cement, and sand [19]. However, this did not produce thin sheet samples either, and due to the complexity of the formula, this approach was not considered in our study.

  12. L229-244: No chronology is needed. Only assumptions, results and discussion.

    12A. Thank you for your suggestion. The whole paragraph is completely written in Lines 220-232. Some of them are repeated here.

    If the load-bearing capacity of 7.5 kg was reached for the sheet of 30×30×2 cm, we then considered the next sheet of size 30×30×1 cm; if this sheet could withstand the load, we considered the next sheet of size 29.0×20.0×0.7 cm. Furthermore, since each sheet of different dimensions needed to be finished and EVA glue applied before a load test of 7.5 kg was conducted at the client’s site, the process would have been time-consuming since the equipment used for glue coating was located at the client’s site, we decide to set a prerequisite for the experiment of 7.5 kg in the client: a load test of 1.5 kg in the university laboratory instead. …

  13.  Fig. 9-11: needed?

    13A. Yes, they are needed. They may be helpful for researchers to duplicate the experiments.

  14. L261 “we adjusted the formula, mainly by increasing the density and adhesion of the cement.”. What does increase adhesion mean? Is adhesion measured/quantified? Specific information would be “the amount of x was increased in the recipe, targeting an increase in cement density and xxx”

    14A. We delete the confusing word adhesion. The new sentence is “If one of these experiments was unsuccessful, we adjusted the formula, mainly by increasing the density of the cement and the amount of seaweed powder.” in Lines 248-250.

  15. Table 1: What is the foaming agent?

    15A. Foaming agent is explained in Lines 280-281. It is repeated here.

    The foaming agent in Step 3 is a liquid, such as hydrogen peroxide and plant surfactants, which can generate foam when mixed with water.

  16. Fig. 17 (c) and 19 (c ) , Fig. 21: Needed?

    16A. Yes, they are needed. They may be helpful for researchers to duplicate the experiments.

  17.  L 413: “without bending” would mean in infinitively stiff plate. Are authors meaning without breaking? 

    17A. Thank you for your suggestion. We changed the confusing word bending to breaking, as shown in Line 404.

I hope the response satisfies you.

Corresponding Author: Jyhjeng Deng    Date: 2023/11/24
